

# Filtering artefacts in bacterial community composition can affect the outcome of dissolved organic matter biolability assays

Joshua F. Dean[1,2], Jurgen R. van Hal[2], Han Dolman[1], Rien Aerts[2], and James T. Weedon[2]

Departments of Earth Sciences[1] and Ecological Sciences[2], Vrije Universiteit Amsterdam, 1081 HV Amsterdam, The
Netherlands.

*Correspondence to*: Joshua F. Dean (j.f.dean@vu.nl)

**Abstract.** Inland waters are large contributors to global carbon dioxide ($CO_2$) emissions, in part due to the vulnerability of dissolved organic matter (DOM) to microbial decomposition and respiration to $CO_2$ during transport through aquatic systems. To assess the degree of this vulnerability, aquatic DOM is often incubated in standardized 'biolability' assays. These assays isolate the dissolved fraction of aquatic OM by size filtration prior to incubation. We test whether this size selection has an impact on the bacterial community composition and the consequent dynamics of DOM degradation using three different filtering strategies: 0.2 μm (filtered-and-inoculated), 0.7 μm (generally the most common DOM filter size) and 106 μm ('unfiltered'). We found that bacterial community composition, based on 16S rRNA amplicon sequencing, was significantly affected by the different filter sizes. At the same time, filtering strategy also affected the DOM degradation dynamics. However, the dynamics of these two responses were decoupled, suggesting that filtration primarily influences biolability assays through bacterial abundance and the presence of their associated predators. By the end of the 41-day incubations all treatments tended to converge on a common total DOM biolability level, with the 0.7 μm filtered incubations reaching this point the quickest. These results suggest that assays to assess the total biolability of aquatic DOM should last long enough to remove filtration artefacts in the microbial population. Filtering strategy should also be taken into account when comparing results across biolability assays.

## 1 Introduction

Research showing that inland waters are significant sources of carbon dioxide ($CO_2$) to the atmosphere (Cole et al., 2007) has led to a large increase in the number of studies that consider the magnitude and source of this $CO_2$. Inland waters are estimated to release $CO_2$ equivalent to ~19% of global anthropogenic $CO_2$ emissions, annually (Le Quéré et al., 2016; Raymond et al., 2013). One potentially important source of this $CO_2$ is dissolved organic matter (DOM) which is present in relatively high concentrations in many inland water systems (e.g. Evans et al. 2014; Dean et al. 2016). The contribution of $CO_2$ from microbial respiration in aquatic systems is an important component of understanding the global carbon cycle (e.g. McCallister and del Giorgio 2012), as well as ecosystem dynamics in this important interface between the terrestrial and marine realms (Aufdenkampe et al., 2011; Guillemette et al., 2017). 'Biolability' assays, which determine the vulnerability





of DOM to microbial decomposition during aquatic transport are an increasingly common approach to determine the magnitude and importance of aquatic DOM as a $CO_2$ source (Guillemette and del Giorgio, 2011; Vonk et al., 2015).

Standardized biolability assays allow for the comparison of DOM vulnerability to decomposition during aquatic transport across a range of systems (Findlay and Sinsabaugh, 2003; Guillemette and del Giorgio, 2011; Vonk et al., 2015). These

approaches involve isolation of the dissolved fraction of organic matter (DOM) over short- (< 2 days) and long-term (~28 days) incubations depending on the research question (Guillemette and del Giorgio, 2011). For determining the total biolability of an aquatic DOM sample, long-term assays tend to be more common (e.g. Spencer et al. 2015).

The DOM size class has variously been defined as the fraction of organic matter molecules smaller than ~1 μm (e.g. del Giorgio and Pace 2008), 0.7 μm (e.g. Mann et al. 2015), 0.45 μm (e.g. Drake et al. 2015), and 0.2 μm (e.g. Logue et al.

2016). The lowest size cutoff, 0.2 μm, is often assumed to be biologically sterile as no microbes are thought to be smaller than this (Gasol and Morán, 1999), although tests have shown that viable microbial communities can develop even in < 0.2 μm filtrate (Hahn, 2004). Filtering to 0.7 μm, arguably the most common size cutoff (the pore size of standard glass fibre filters used in water filtering applications – GF/F) would also likely exclude the majority of microbes (Ferguson et al., 1984; Gasol and Morán, 1999). Biolability assays that filter to 0.2 μm commonly include an inoculation of the incubations after

filtration with in-situ (or study relevant) unfiltered microbial communities to avoid community effects (e.g. Logue et al. 2016). However, biolability assays filtered to 0.45 and 0.7 μm are often not inoculated and it is assumed that enough bacteria will pass through the filter to decompose and respire the filtrate DOM (Vonk et al., 2015). In biolability studies specifically in permafrost regions, the effect of filter size was shown to be insignificant (Vonk et al., 2015). However, due to the extreme size filtration this is a surprising result given that these filter sizes will exclude such a large proportion of microbes,

potentially causing a significant shift in the microbial community structure and therefore potentially altering the dynamics of the decomposition processes mediated by the microorganisms (Logue et al., 2016; Traving et al., 2016).

We aimed to test the significance of this potential microbial shift resulting from filtration by answering the following questions: (1) How do different filtering strategies affect the bacterial community composition in DOM incubations? (2) Does this influence the outcome of DOM biolability assays? We tested these questions using an experimental setup with

three filtration treatments applied during biolability assays of organic carbon-rich water draining a temperate peatland in the Netherlands.

## 2 Methodology

### 2.1 Field Sampling

The organic carbon-rich water for the DOM incubations was collected from a ditch draining Horstermeer peatland (52.144°

N, 5.043°E) (Hendriks et al., 2007). The site is former agricultural land, abandoned in the mid-1990's, in a drained natural lake in the central Netherlands. The underlying geology is Pleistocene eolian sands, overlain by peat and organic-rich lake deposits (Hendriks et al., 2007). Approximately 20 L of water was collected in Spring 2016 (16 March), pre-filtered to



106 μm, and transported to Vrije Universiteit Amsterdam in two acid-washed plastic vessels and refrigerated until filtered for the incubations on the same day.

## 2.2 Incubations

The sample water was prepared for incubation with three filtering strategies: (1) 3 L of sample water filtered to 0.2 μm (sterilized nylon membrane, Whatman), homogenized and inoculated with 5 ml of sample water filtered to 106 μm (henceforth P2 treatment). (2) 3 L of sample water filtered to 0.7 μm (pre-ashed glass fibre filter, Whatman; GF/F) and homogenized (henceforth P7 treatment). (3) 3 L of sample water homogenized but not filtered further than 106 μm (henceforth UF – unfiltered). Sub-samples for DOM carbon concentrations and quality analyses (CDOM) were collected from the homogenized treatments.

For each treatment, 100 ml of sample water was decanted into 25 individual 125 ml acid-washed HDPE bottles for incubation. The bottles were incubated at 20°C in completely dark conditions following standard protocol (Vonk et al., 2015). Bottle lids were loosely placed on top to ensure there was sufficient oxygen for bacterial respiration to occur. Five bottles for each treatment were removed from the incubation chamber on days 5, 14, 27 and 41 to enable sampling and analysis of DOM and microbial community dynamics during incubation. All incubation bottles were gently shaken every 2 to 3 days to limit flocculation and settling of organic matter at the bottom of the incubation bottles.

Five individual 125 ml acid-washed HDPE bottles were filled with 100 ml demineralized water and incubated under the same conditions as the treatments to serve as blanks. Carbon concentrations in the blanks increased from 0.01 ± 0.02 to 0.05 ± 0.02 mmol/L over 41 days representing an increase of 0.03 mmol/L, likely because the demineralized water was not completely sterile. These values represent ~1% of the initial DOM carbon concentrations and a maximum of ~15% of the lowest final DOM carbon concentration suggesting that any possible carbon input from the plastic bottles was small relative to the DOM dynamics observed during the incubation. Incubation samples would not have been affected by the suspected biological growth in the non-sterile demineralized blanks because the incubation bottles only contained sample water and were not treated with the demineralized water in any way.

## 2.3 Sample collection and analysis

DOM carbon concentration sub-samples were collected by re-filtering to isolate the dissolved component (Vonk et al., 2015). For the P2 treatment, these sub-samples were filtered to 0.2 μm (regenerated cellulose membrane, Whatman) using sterile syringes; for the P7 and UF treatments, they were filtered to 0.7 μm (pre-ashed GF/F, Whatman) – this was to ensure the same size class of DOM was compared for each treatment. Samples were frozen immediately after collection and analyzed within 6 months. DOM carbon concentrations were measured on a high-temperature combustion total organic carbon system (varioTOC cube; Elementar Analysensysteme GmbH) coupled to an isotope ratio mass spectrometer (IsoPrime 100; Isoprime Ltd, UK), using an independent calibration curve (measurement range 0.02 to 12.5 mmol/L) (Federherr et al., 2014; Kirkels et al., 2014).



DOM absorbance measurements (CDOM) were carried out immediately after sample filtration using a double-beam spectrophotometer (Shimadzu UV-1601PC). Absorbance spectra were determined in matched 1 cm quartz cuvettes over 200-750 nm, at 0.2 nm intervals, using fresh demineralized water as reference. We calculated standard CDOM indices: $SUVA_{254}$, $S_R$, E2:E4 and E4:E6 ratios, and also present the specific absorption (a) at 240, 300, and 440 nm (Helms et al., 2008). Measured absorbance values were converted to absorption coefficients:

$$a_{CDOM}(\lambda) = 2.3 \times A(\lambda)/l \qquad (1)$$

where $A(\lambda)$ is the absorbance at wavelength $\lambda$ (nm) and l is cuvette path length (m). The slope coefficient, $S_{CDOM}$ in $nm^{-1}$, is defined in the following equation (Helms et al., 2008):

$$a_{CDOM}(\lambda) = a_{CDOM}(\lambda_r)e^{-S_{CDOM}(\lambda-\lambda_r)} + K \qquad (2)$$

where $a_{CDOM}(\lambda_r)$ is the CDOM absorption at a reference wavelength $\lambda_r$, and $K$ is the parameter to offset baseline shifts unrelated to CDOM absorption. $S_{CDOM}$ was fitted for each sample over the wavelength ranges of 275-295 nm and 350-400 nm using linear regression of the log-transformed absorption coefficients. The ratio of these spectral slopes ($S_R$) has been shown to provide insights into changes in DOM molecular weight (Helms et al., 2008).

**2.4 DNA extraction and amplicon sequencing**

At each sampling occasion, 20 mL of each sample were filtered to 0.2 μm (regenerated cellulose membrane, Whatman) using sterile syringes. These filters were subjected to DNA isolation for subsequent 16S amplicon community profiling. DNA was extracted from filters using the MoBio Powersoil DNA isolation kit (MoBio, Carlsbad, USA) following the standard protocol. Successful DNA extraction was confirmed by visualization on agarose gel and quantification with NanoDrop (ThermoFisher, USA) in comparison to procedural blanks (sterile filters). Amplicons for sequencing were generated by a two-step PCR. The first PCR was with universal bacterial and archaeal primers targeting the V3-V4 region of the 16S rRNA gene (341F and 806R). Reaction mixtures were 12.5 μL Phusion High-Fidelity Mastermix (ThermoFisher Scientific, USA), 1 μL of each primer at 10 nM concentration, 9 μL of nuclease free water and 1.5 μL of DNA sample. The PCR program was: initial step at 98°C for 30 s, 30 cycles of: 98°C for 30 s, 55°C for 30 s, 72°C for 30 s; and final step of 72°C for 10 min. After confirming successful amplification, PCR products were purified and normalized using Sequalprep plates (Thermofisher, USA), and subject to a second indexing PCR such that each sample received a unique combination of 6-nucleotide barcoded forward and reverse primers. The reaction mixture was as above, and the PCR program was initial step at 95°C for 30 s, 8 cycles of: 95°C for 30 s, 55°C for 30 s, 72°C for 30 s; and final step of 72°C for 5 min. PCR products were again purified and normalized with Sequalprep, then pooled, gel purified with QIAquick Gel Extraction kit (Qiagen, USA), quantified with qPCR and the KAPA Library Quantification kit, and sequenced on an Illumina MiSeq, with V3 chemistry, 2 x 300 cycles, and a target PhiX concentration of 20%.



## 2.5 Bioinformatics

Our amplicon sequencing procedure produced 4.9 million pairs of raw reads. The original intention was to assemble contigs from the paired-end reads, but low read quality in the tail (last 150 cycles) made merging of paired-end reads impossible. We therefore analyzed single reads trimmed to 180 and 150 bases for forward and reverse reads respectively. We performed analysis on both forward (V3 region) and reverse (V4 region) reads in parallel using a common bioinformatics pipeline.

Raw reads were processed by trimming primers and applying quality filtering with the recommended maximum expected error rate of 1 (Edgar and Flyvbjerg, 2015). Unique reads were subsequently sorted by abundance, singletons discarded and OTUs defined using the UPARSE-OTU algorithm (Edgar, 2013) with minimum similarity set at 97%. The original set of truncated and quality-filtered reads were mapped back to the resulting OTUs to create an OTU table of read abundance per OTU per sample. Representative sequences for each OTU were aligned using PyNAST (Caporaso et al., 2010a) and assigned to taxonomy using the RDP classifier (Wang et al., 2007) using QIIME version 1.7.0 scripts (Caporaso et al., 2010b) and Greengenes 2011 release as the reference database (DeSantis et al., 2006). Candidate OTUs that failed to align with more than 70% similarity were discarded as putative artifacts. All remaining OTUs were assembled into a phylogenetic tree using the FastTree algorithm (Price et al., 2009). Subsequent statistical analyses were performed on OTU tables rarefied to 8000 sequences per sample.

## 2.6 Statistics

The DOM carbon concentration measurements exhibited complex temporal dynamics that differed between filtering treatment making it impractical to fit a single parametric model, so we analyzed the data using Generalized Additive Models (GAMs) (Wood, 2006). We fit a series of models, in every case modeling DOM carbon concentration as a smooth function of time, but by either fitting (1) a single smoother for all treatments combined, (2) a separate smoother for each treatment, or (3) a single smoother for P2 and UF and a separate one for P7. For each model we also tested the effect of choice of smoothing parameter ($k = 3$ or 4, related to the degree of flexibility permitted in the curve fitting). Model selection using Akaike information criterion (AIC) (Johnson and Omland, 2004) was employed to identify the best-fitting model from these six candidate models.

For each of the seven DOM quality indices, we analyzed the effects of incubation time and treatment, and their interaction, using a two-way ANOVA model. Since each observation was from an independent incubation flask, no adjustments for repeated measurements were made. Time was treated as a discrete factor.

Relationships between bacterial community profiles were visualized by Nonmetric Multidimensional Scaling (NMDS) ordination of Unifrac distance matrices computed from the rarefied OTU table and associated *de novo* tree of OTU representative sequences. Unifrac is a pairwise phylogenetic similarity measure defined as the proportion of tree length shared by OTUs detected in any two samples (Lozupone and Knight, 2005). It therefore expresses the degree of similarity between samples, weighted by the phylogenetic distinctness of the differentiating taxa. We performed ordinations based on



both the *unweighted* (taking into account only presence or absence of taxa) and *weighted* (also weighing taxa by their abundances in the OTU table) versions of Unifrac, as these provide complementary explorations of the relationships between community profiles (Lozupone and Knight, 2005). More formal analysis of the difference in community profiles between treatments and sampling occasions was performed using permutational multiple analysis of variance (PERMANOVA)

(Anderson, 2001; Oksanen et al., 2011), using both the weighted and unweighted Unifrac matrices as the response variable, and filtering treatment and sampling time as fully crossed fixed factors. All statistical analyses were performed using the R software package (R Core Team, version 3.4.4).

## 3 Results

### 3.1 DOM carbon dynamics

DOM carbon concentrations in sample water measured directly after field collection averaged 1.09 mmol/L (Standard Error [SE] = 0.01, $n$ = 15; Figure 1). Over the course of the incubation, there was a decrease in DOM carbon concentrations with different dynamics between filtering strategies. P7 showed roughly negative exponential decay dynamics from the beginning (Figure 1) and reached a final carbon concentration after 41 days of 0.225 mmol/L (SE = 0.001, $n$ = 5) equivalent to 21% of initial concentrations. Both UF and P2 showed temporal dynamics with a clear 'lag-phase': large decreases in DOM carbon

concentrations were not apparent after 28 days, but after 41 days they approached convergence with P7, with final values at 49% and 31% of initial concentrations, respectively (Figure 1). This difference in dynamics according to filtering strategy was statistically significant when analyzed with GAMs, since the best fitting model included separate smoothers for each filtering treatment (*adjusted $R^2$* = 0.89, estimated *d.f.* = 11.0, Table 1). Two points at UF day 41 could be considered outliers; however, removing them from the dataset leads to qualitatively the same conclusions with regards to the best fitting model

by the AIC model comparison, and the following predicted values at day 27 (proportion DOM carbon remaining, mean ± standard error, $n$ = 5 in all cases; P2 = 98% ± 3, P7 = 27% ± 3, UF = 86% ± 3), and at day 41 (P2 = 31% ± 3, P7 = 21% ± 3, UF = 24% ± 4) (Table 1; Figure 1).

DOM carbon concentrations in P2 and P7 were slightly higher than initial measurements on day 5 (Figure 1), suggesting there may have been growth of chemoautotrophic organisms (photoautotrophy can be ruled out given the dark incubations).

A likely candidate is nitrification, given the relatively large concentrations of ammonia in the water column (29-82 μmol/L $NO_3 + NO_2$, J. van Huissteden, pers. comm.). This is supported by rapid growth in microbial cell counts over the same period (Logue et al., 2016) (Supplementary Figure S1).

DOM quality showed a consistent pattern across filtering treatments. All the absorption parameters indicate a rapid shift from more aromatic/higher molecular weight DOM to much more degraded and smaller molecules by day 5 (Figure 2). This

is more pronounced in SUVA254, E2:E3 ratios and absorptions at 240, 300 and 440 nm compared to $S_R$, the latter of which shows greater scatter but the same general trend. Moreover, factorial ANOVA analyses show that the temporal effect was always stronger than any effect of filtering treatment (Table 2). The E4:E6 ratios did not change significantly over the course




of the experiment, nor did they differ between filtering strategy treatments. This shows that the DOM carbon concentration dynamics are decoupled from DOM structural dynamics, indicating that microbial activity is playing an important role in carbon concentrations but that overall molecular degradation was consistent across the treatments.

### 3.2 Microbial community structure

Bacterial community analyses based on the V3 and V4 regions were qualitatively very similar. We here present results based on the longer and higher quality V3 sequences. Figure 3a,b displays the relationships between bacterial community profiles based on treatment and sampling time. There is a clear temporal shift in bacterial community composition, the trajectory of which varies due to filtering treatment. This interpretation is supported by the results of the PERMANOVA, which show that for the abundance-weighted analysis both incubation time, treatment and their interaction significantly contribute to variation

in the community distance matrix (all permutation $P < 0.0001$, proportion of variance explained: time = 66%, filtration treatment = 8%, time x filtration = 15%). For presence-absence based analysis, treatment and time effects were still significant but the latter was relatively weaker (all permutation $P < 0.0001$, proportion of variance explained: time = 29%, filtration treatment = 12%, time x filtration = 14%).

The patterns described above are a result of complex temporal dynamics in the relative abundances of different bacterial taxa

(Supplementary Figures S2 & S3). Focusing on the most abundant taxa, across all treatments the relative abundances of reads assignable to *Alphaproteobacteria, Bacteroidetes, Verrucomicrobia, Planctomycetes* and *Actinobacteria* all tended to increase over the course of the incubation. Conversely, reads assignable to the candidate phylum *OD1* and *Parvarchaeota,* and unassignable reads, all showed negative trends in relative abundance over the course of the incubation. Reads assignable to the classes *Beta-, Gamma-, Epsilon-* and *Gammaproteobacteria* showed more variable dynamics exhibiting large changes

in relative abundance in the first 5 days, followed by stabilization for the remainder of the incubation. When comparing treatments, the major differences were due to the timing and magnitude of shifts in these general patterns. More specifically, *Actinobacteria* showed a much stronger increase in relative abundance over the incubation in UF samples relative to the other treatments; *Bacteroidetes* started lower and peaked later in P7 relative to the other two treatments; *Betaproteobacteria* had a less pronounced peak in UF relative to the other two treatments; UF samples showed relative lower

*Alphaproteobacteria* and higher *Deltaproteobacteria* relative abundances in the latter part of the incubation; *Verrucomicrobia* reached a higher relative abundance in P2 samples relative to the other two treatments; for *OD1* and *Parvarchaeota,* P7 had higher initial relative abundances than the other two treatments, but for both taxa the abundances from 14 days onwards were broadly similar across treatments (Figure 3; Supplementary Figures S2 & S3).



## 4 Discussion

Filtering prior to incubation is standard practice in DOM biolability assays. Our results show that the specific choice of filtering strategy can have persistent consequences for the community composition of bacterioplankton throughout the course of the incubation, as well as on the dynamics of DOM degradation.

### 4.1 How does filter treatment affect bacterial community composition?

The community profile data show consistent succession in bacterial community composition over the 41 days of incubation (Figure 3). Similar temporal patterns of succession over short time scales are regularly observed in bottle incubations (e.g. Massana et al. 2001; Baltar et al. 2012), and are also characteristic of seasonal dynamics in bacterioplankton communities *in-situ* (Gilbert et al., 2009; Rösel et al., 2012). These changes most likely reflect dynamics driven by different growth rates, resource competition and, potentially, predation.

In our study system, UF samples (unfiltered) and P2 samples (filtered to 0.2 μm and inoculated with unfiltered sample) had similar bacterioplankton community compositions at the beginning of the incubation. During the course of the incubation, however, the bacterioplankton community compositions of P2 and P7 (filtered to 0.7 μm but not inoculated) treatments became more similar, a pattern that persisted until the end of the incubation. When assessed with a presence-absence based distance metric, the different treatments did not converge (Figure 3a), showing that for the timescales relevant for incubation, filtering strategy significantly alters bacterioplankton community composition.

Filtering strategy is likely to have affected the bacterial community composition in our incubations through three different mechanisms.

Firstly, the direct size selection effect of the different filter treatments is likely to have enriched for bacterial taxa that can pass through a given filter size (Gasol and Morán, 1999), thereby determining the community structure at the beginning of the incubation. In our study, samples filtered to 0.7 μm were particularly enriched in candidate phylum *OD1* and *Parvarcheota*, but depleted in *Bacteroidetes*, *Epsilon-*, *Delta-* and *Betaproteobacteria* at the very beginning of the incubation relative to P2 and UF samples. *OD1* is a poorly characterized taxon, although some evidence points to its members being particularly abundant in the hypolimnion of lakes (Peura et al., 2012). *Parvarchaeota* are considered amongst the smallest microorganisms currently known (Chen et al., 2018), so their higher relative abundance is not surprising given they would readily fit through the 0.7 μm filter pore. The various *Proteobacteria* classes encompass a wide range of ecological niches in freshwater environments (Newton et al., 2011). This wide variety makes it difficult to relate the observed patterns in our study to specific functions, but the depletion of *Betaproteobacteria* at the beginning of the incubation in P7 samples agrees with the observation that members of this group tend to be fast-growing and therefore tend to have larger cell sizes (Newton et al., 2011). Of the other two treatments, it is interesting to note that the P2 samples more closely resembled the UF samples in the early stages of the incubation. This implies that, initially at least, the bacterial community in P2 was dominated by taxa




introduced with the inoculum, despite the fact that members of some freshwater bacterial phyla can pass a 0.2 μm filter (Hahn, 2004). This is evidence for the efficacy of inoculation treatments in biolability studies.

A second mechanism by which filtering could affect bacterial community composition is through the activity of bacterial grazers. Given the important role of bacterial predation by ciliates, flagellates and other microeukaryotes in structuring

bacterioplankton communities (Hahn and Hofle, 2001), it is logical to expect that any filtering strategy that reduces their abundance would influence the development of the bacterioplankton community in an incubation (although we did not characterize the bacterivorous community in this study). For example, if UF samples contained a bacteriovore community closest to that found in the source system, then the higher relative abundances of *Actinobacteria* and lower peak of *Betaproteobacteria* could result from grazing-mediated selection; *Actinobacteria* are generally considered defense

specialists, while *Betaproteobacteria* have been observed to be sensitive to grazing pressure (Hahn and Hofle, 2001; Newton et al., 2011).

A third possible mechanism by which filtration could influence bacterial community composition is the effect of filtering on the quantity and chemical composition of the DOM pool. If DOM chemical composition is related to its size fraction, then filtration could potentially change the quality/structure of the substrate available to heterotrophs leading to divergences in

community composition due to differential specialization on carbon substrates (Logue et al., 2016). The different filtration strategies used in this study did not significantly affect the overall structural characteristics of the DOM pool (Table 2; Figure 2), so this is unlikely to have influenced the bacterial community composition or its evolution through time. Further, the bacterial community was observed to shift over longer timescales than any shift seen in the DOM structural properties, which all reached equilibrium by day 14 (Table 2; Figure 2).

**4.2 Does the change in bacterial community composition affect DOM biolability assays?**

The DOM dynamics of P7 were distinct from P2 and UF, which in general behaved the same through the course of the incubation (Figure 1). This is the opposite to what we see in the bacterial community composition dynamics in which UF was distinct from P7 and P2 (Figure 3). DOM degradation dynamics may therefore be driven by the bacterial community composition commonalities in P2 and UF, and that the bacterial community in P7 (the most common treatment/experimental

design for DOM biolability studies) (Vonk et al., 2015), is more efficient at degrading DOM. However, in this study there was no evidence for the presence of high-level taxa that clearly associate with variations in DOM dynamics. This is most likely due to the high diversity of ecological traits within the higher level taxonomic groupings we analyzed (Martiny et al., 2015). All the major groups observed, especially *Actinobacteria*, *Bacteriodetes* and *Proteobacteria*, are known to be associated with the degradation of DOM in freshwater systems (Bauer et al., 2006; Gattuso, 2002; Newton et al., 2011). The

lack of clear connection between these taxa and observed DOM degradation leads to the conclusion that degradation was not driven by specific changes in the bacterial community composition. We propose that the following two mechanisms could explain the observed dynamics:





(1) Bacteriovores (grazers) limit the growth rate of bacterial communities (Baumgartner et al., 2016; Hahn and Hofle, 2001). This could explain the difference between P7 and UF, as the 0.7 μm filter pore size would likely exclude bacteriovores, which tend to be much larger than the bacteria they prey upon (Berdjeb et al., 2011). However, bacteriovores are likely to be present in the UF and P2 treatments (in the latter case due to inoculation).

(2) Filtration will severely limit the number of cells that pass through specific filter pore sizes (Wang et al., 2008). In P2, DOM degradation rates may have been limited by the low initial population size caused by the size cut off of the 0.2 μm filter, which would exclude the vast majority of bacteria (Hahn, 2004). The number of bacterial cells in the P2 treatments were consistently 31-75% lower than in the UF treatments (Supplementary Figure S1).

In this system, there is no evidence that filtering effects on DOM structure can explain the patterns in DOM degradation, nor

(as discussed above) the bacterial community dynamics. We argue that filtration will instead influence the DOM dynamics mainly by a combination of influences on bacterial community composition, bacterial abundance and the presence of their associated predators.

## 4.3 Limitations and future work

This study is from a single peatland site in the Netherlands and therefore may not be representative of DOM biolability

dynamics across a wider range of freshwater ecosystems. The organic carbon-rich nature of the study site and its water are consistent with temperate peatland characteristics elsewhere in the Northern Hemisphere (e.g. Billett et al. 2010), but the agricultural history of the site means that nutrient availability and the general chemistry of the sample waters are likely different from more pristine peatland systems. Such environmental factors can influence microbial communities (Dean et al., 2018; Gulis and Suberkropp, 2003), which would in turn have influenced the development of bacterial community

composition during the experiment presented here.

It is important to note that the microbial community patterns described here are based on relative abundances; thus, apparent declines of particular taxa may be attributable to either real declines in population, or (as is more likely) relatively slower growth relative to other taxa in the community. Moreover, PCR amplicon generation is known to introduce biases in the community profile (e.g. Sipos et al. 2007), so any interpretation of these data for particular taxa should be further confirmed

by more reliable quantitative methods such as taxon-specific qPCR (Smith and Osborn, 2009), FISH (Amann et al., 1995) or PCR-free metagenomics methods (Handelsman, 2004).

Despite the lack of clear connection between bacterial community composition and DOM dynamics, the high reproducibility of both measurements across technical replicates suggest that this system could be a potentially useful model for further exploring the relationship between community structure and function. A central question in microbial ecology is to what

extent information about the taxonomic composition of microbial assemblages allows prediction of their biogeochemical function (Bier et al., 2015; Graham et al., 2016). Future studies should seek to experimentally test the relative importance of initial population size and the presence or absence of grazers in determining DOM utilization rates. Future work should also consider bacterial community composition dynamics, size selection and DOM structural characteristics across multiple sites



and climate settings to confirm the degree to which filtration strategy affects DOM biolability assays across a wider range of catchment settings.

## 5 Conclusions

Filtering strategy was shown in this study to affect both microbial community composition and DOM degradation dynamics, but these two responses were disconnected from one another. There are two important conclusions from these results for interpreting aquatic biolability assays. Firstly, our results suggest that care should be taken when comparing results across different filtering strategies, especially for shorter incubations, since we have shown that the DOM degradation and microbial community dynamics may not converge until after the commonly used 28-day length for long-term biolability assays, if at all (Figures 1 and 3). However, in many cases the relevant parameter to be estimated from such long-term incubations is the total 'biolabile fraction' (Guillemette and del Giorgio, 2011), and despite the divergent dynamics in the early part of the incubation it appears that all filtering strategies do eventually converge on a similar value for total DOM biolability. The P7 (0.7 μm filtration with no inoculum) treatment appeared to reach this point at the fastest rate.

## Acknowledgements

This work was carried out under the program of the Netherlands Earth System Science Centre (NESSC), financially supported by the Ministry of Education, Culture and Science (OCW) (grant number 024.002.001). We thank the Systems Bioinformatics group at VU Amsterdam for the use of their flow cytometry equipment. The authors declare that they have no conflict of interest.

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

5 **Tables**

**Table 1: Analysis of temporal DOM carbon concentration dynamics. Three candidate GAM models were fit including a non-linear smoothing term *s*. These models correspond to the scenarios (a) treatments differ in initial concentration but show identical temporal dynamics; (b) treatments differ in initial concentration but with unique temporal dynamics of DOM carbon** 10 **concentration for each treatment; (c) treatments differ in initial concentration but with separate dynamics for P7 relative to P2 and UF. For each model two possible fitting parameters (*k*) were tested. Higher values of *k* allow more flexible model forms. Values reported are AIC with the % deviance explained by the model in parentheses, followed by the estimated degrees of freedom. AIC is a measure of relative goodness of fit for each model to the observed data; lower AIC values indicate better fit. The best fit model is printed in bold and was used to generate the non-linear regression lines in Figure 1.**

|  | *Fitting parameter* | |
|---|---|---|
| **Explanatory variables** | *k* = 3 | *k* = 4 |
| a) Treatment + *s*(Day) | -33 <br><br> (73.7%, *df* = 4.9) | -33 <br><br> (74.2%, *df* = 5.4) |
| b) Treatment + *s*(Day, by = Treatment) <br><br> *Treatment: P2 vs P7 vs UF* | -60 <br><br> (83.1%, *df* = 7.8) | **-97** <br><br> **(90.4%, *df* = 11.0)** |
| c) Treatment* + *s*(Day, by = Treatment*) <br><br> *Treatment*: [P2, UF] vs P7* | -54 <br><br> (80%, *df* = 5.0) | -82 <br><br> (87.3%, *df* = 7.7) |





**Table 2: Analyses of DOM spectral properties, with ANOVA values for each of the DOM spectral indices. Each row represents a different index. Each column indicates the F ratio for the corresponding term in a two-way factorial ANOVA model. Numerator and denominator degrees of freedom are given for each column. Within a row, larger values indicate relatively larger effects. Asterisks indicate significance levels (* $P < 0.05$, ** $P < 0.01$, *** $P < 0.001$).**

|  | Day $(d.f. = 4,58)$ | Treatment $(d.f. = 2,58)$ | Day x Treatment $(d.f. = 8,58)$ | Adjusted $R^2$ |
|---|---|---|---|---|
| **Slope Ratio ($S_R$)** | 52.7 *** | 9.5 ** | 2.3 * | 0.76 |
| **SUVA$_{254}$** | 1679.7 *** | 0.1 | 7.4 *** | 0.99 |
| **Absorbance 240 nm** | 1774.6 *** | 0.2 | 6.4 *** | 0.99 |
| **Absorbance 300 nm** | 1512.2 *** | 2.1 | 10.9 *** | 0.99 |
| **Absorbance 440 nm** | 546.5 *** | 36.7 *** | 32.9 *** | 0.97 |
| **E2:E3** | 103.7 *** | 2.6 | 1.8 | 0.85 |
| **E4:E6** | 2.8* | 0.6 | 1.4 | 0.12 |





**Figures**

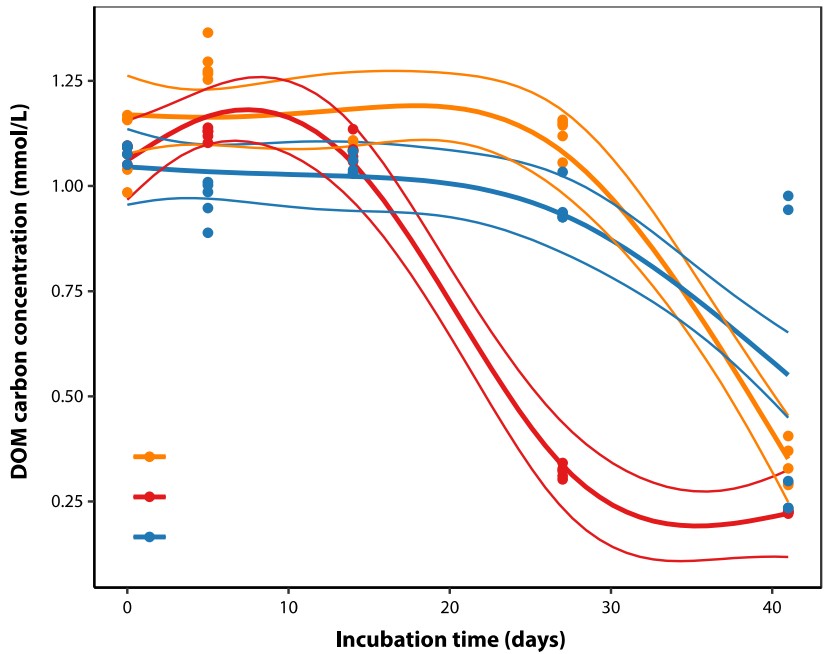

**Figure 1: DOM carbon concentration dynamics over the course of the incubations for the three filtration treatments. The thicker smoothed lines represent the best-fit GAM as selected using AIC (see Section 2.6), thinner lines define the bounds of the 95% prediction intervals derived from the model fits, with standard error estimates pooled across all observations (Table 1).**





**Figure 2: DOM quality (structural proxies) dynamics during the course of the incubations for each filtration treatment, separated into the seven different DOM structural indices used in this study (a-g; see Section 2.3). Points represent individual replicates, lines connect mean values calculated for each "Day x Treatment" combination (Table 2).**





**Figure 3: Bacterial dynamics for each of the filter treatments during the incubations. a) & b) NMDS ordinations based on presence-absence (a) or abundance-weighted (b) UniFrac distances; clusters (convex hulls) for each day x treatment combination are plotted, the lines join the center of each cluster following the sequence of sampling. c) relative read abundances of the nine most abundant bacterial (sub-) phyla, based on 16S rRNA gene amplicon sequence data, separated by sampling time and treatment. Each column represents an experimental replicate. "Other" contains both unassigned Bacterial reads and reads assigned to less abundant taxa. See supplementary material for univariate time series plots for the most abundant taxonomic groups.**