# Peer review of "Filtration artefacts in bacterial community composition can affect the outcome of dissolved organic matter biolability assays"

_Biogeosciences, 2018_

## Referee Comment (RC1) · Anonymous Referee #1 · 8 Jul 2018

This is overall a short but relevant manuscript, that provides experimental data on DOM degradation of surface waters under different filtration conditions.

-"DOM carbon concentrations" is odd and confusing – I suggest to use "DOC concentrations"

-The description of DOC temporal variations in the experiments (Figure 1) is perhaps not very convincing. Based on many published degradation experiments, one would expect to see exponential losses, the smoothing approach here is odd.

-DOC concentrations were measured using a TOC-IRMS setup, it is unfortunate that d13C data are not presented, these could have a strong added value to the manuscript

and would allow to expand the discussion with other published studies looking at changes in d13C during microbial degradation, or to shed some light on the suggestions that an initial (small) increase in DOC could be due to nitrification.

-It would be good to include some discussion/references on how the filter pore size might (or might not) affect the DOC concentrations and characteristics. As a starter, both papers below demonstrate no significant differences in DOC and d13C-DOC between 0.2 and 0.7 $\mu$m filtrations: Denis et al. (2017) A comparative study on the pore-size and filter type effect on the molecular composition of soil and stream dissolved organic matter. Organic Geochemistry 110: 36–44 Bouillon et al. (2014) Contrasting biogeochemical characteristics of the Oubangui River and tributaries (Congo River basin). Scientific Reports 4 : 5402 | DOI: 10.1038/srep05402

-throughout the ms, you refer to either 'filtering' or 'filtration' – I would prefer to stick to the latter

-Figure 1: I do not see a legend to indicate which experiments the data correspond to.

---

## Author Comment (AC1) · 23 Jul 2018

We thank the referee for the positive comments. Below we give a first response to each of the points raised initially. To facilitate discussion, we also give a more detailed response to the two most important issues in order to expand on the points the referee raises (denoted with *). We will post a final response and modified manuscript after the Open Discussion is closed.

- We used "DOM carbon concentrations" so we didn't confuse readers by switching between the acronyms "DOM" and "DOC", but we acknowledge this is not the norm so we will revert to using both acronyms.

[Figure]

\* - The reviewer is correct that it is more common to fit a negative exponential model to data of this type. However, this function is constrained to be strictly decreasing and therefore cannot be sensibly fit to data that show increasing values with time, or even lag-phase dynamics. Rather than impose a functional form on the data we therefore decided to use a smoothing approach (GAMs), combined with model selection methods to guard against overfitting. This approach has been successfully applied elsewhere (Catalan et al. 2017, JRG:Biogeosciences, DOI:10.1002/2016JG003512). We have also fit negative exponential models to the same data (resulting in much poorer fits, AIC = -13, see attached Fig. 1), and can include this and supply the details and results as Supplementary Material if required.

\* - We initially excluded these results because there was very little change in the d13C values during the experiment. However, after exploring the literature further, and after the encouragement of the reviewer, we found that reporting d13C-values during these incubations is rare. We now include these results in the attached Fig. 2, and below we discuss the results briefly. This figure can be included in the manuscript and the discussion expanded to cover this data if the referee(s) and editor agree that it adds value:

The d13C-DOC values remain stable during incubation, within $\pm$ 2.0 ‰ (Fig. 2), although this variability is slightly more pronounced than in other dark bacterial decomposition experiments of aquatic DOC ( $\pm$ 0.0-0.5 ‰ ) (Lalonde et al. 2014, Biogeosciences, DOI:10.5194/bg-11-3707-2014; Vahatalo and Wetzel 2008, Limnol. Oceanogr., DOI:10.4319/lo.2008.53.4.1387). The shifts in d13C-DOC could be due to the preferential mineralisation of certain DOC molecules with distinct d13C signatures - a more positive d13C signature for all samples at day 5, and a more negative d13C signature in only the P7 samples at day 14. The former shift reflects the initial increase in DOC concentrations seen in the P2 and P7 treatments, and the latter shift reflects the change in DOC degradation dynamics seen only in the P7 treatment. However, these shifts in d13C-DOC do not provide any clear line of reasoning for these differing

DOC dynamics (for example a shift towards less negative d13C-DOC values due to chemoautotrophic fixation of ambient CO2 at day 5). As the experiment progressed, the d13C across all samples converged, although over a range of 1 ‰ ( -28.0 to -29.0 ‰ ).

- We will add a short discussion on the expected influence of filter pore size on DOC concentration and characteristics starting with the references as suggested.

- We will ensure consistency throughout the ms with reference to "filtration"

- We will correct the missing labels in Figure 1.
* * *
[Figure]

**Fig. 1.**

[Figure]

**Fig. 2.**

---

## Referee Comment (RC2) · Anonymous Referee #2 · 15 Aug 2018

Dean et al. present a concise method focused manuscript into the use of different filter (pore) sizes and the consequences of these on DOM degradation. The manuscript is well written and organised and easy to follow. I have the following comments:

Methodology

Line 33- What temperature were the samples refrigerated at, would be useful to note? And were the samples cold stored on the journey back from the field to the University.

Was there a reason that only sizes 0.2 and 0.7$\mu$m pore sizes were chosen? As 0.45$\mu$m is also a very common size, particularly in tropical peatland studies.

[Figure]

Just wondered why it took 6 months to analyse the samples for their carbon concentration?

Results

Figure 1. There is no indication of the legend to highlight which line/ colour refers to P2, P7 and UF.

Figure 2. Perhaps the marker points could be made a bit smaller, so it is easier to distinguish the individual points.

Include P2, P7 and UF under the Day 0 and Day 14 in Figure 3c.

Discussion

It would be good to see more of a discussion concerning the impact of pore size on DOC concentration, and particularly how this may change over time. For example, would any pore size be fine if you are going to analyse the sample quickly?

---

## Author Comment (AC2) · 30 Aug 2018

We thank the reviewer for their constructive comments on our manuscript. Here we give our initial response to these comments, and will provide a modified manuscript after the discussion is closed and under the guidance of the editor. We also include a response to both reviewers at the end of this comment as they both raised the same point regarding expanding the discussion about the impact of filtration on DOC structure and concentrations.

Methodology:

- Line 33: The samples were refrigerated at 4 degC. The samples were not refrigerated during transport, but the travel time was less than an hour and were refrigerated immediately upon arrival at the university. These notes will be added to the methodology in our modified manuscript.

- We chose 0.2 and 0.7 $\mu$m pore sizes as these are the most commonly used filter sizes for DOM incubations and provide a reasonable spread in pore size. The reviewer is correct in that 0.45 $\mu$m is common as a size definition or filtration choice for normal DOC concentration analyses (as mentioned in our introduction, pg 2 line 9). But 0.45 $\mu$m is not commonly used for incubations unless an inoculum is added, in which case we would consider a 0.45 $\mu$m treatment to be analogous to our 0.2 $\mu$m treatment; we did not have the budget for a fourth treatment in this experiment.

- It took 6 months to analyse the samples because our own lab for running DOC concentrations had technical problems, so we had to find another lab to run these samples. This took some time and there was a wait to finally run the samples. 6 months frozen storage should not have had any significant impact on the concentration and isotopic composition of the DOC samples (Gulliver et al., 2010; Peacock et al., 2015).

Results:

- Figure 1. We will correct the missing labels.

- Figure 2. We will shrink the marker points as requested.

- Figure 3c. We will include the P2, P7 and UF labels under Day 0 and Day 14.

Discussion:

Reviewer 2 states: "It would be good to see more of a discussion concerning the impact of pore size on DOC concentration, and particularly how this may change over time. For example, would any pore size be fine if you are going to analyse the sample quickly?"

This comment is echoed by Reviewer 1, who states: "It would be good to include some discussion/references on how the filter pore size might (or might not) affect the DOC concentrations and characteristics. As a starter, both papers below demonstrate no significant differences in DOC and d13C-DOC be- tween 0.2 and 0.7 $\mu$m filtrations: Denis et al. (2017) A comparative study on the pore- size and filter type effect on the molecular composition of soil and stream dissolved organic matter. Organic Geochemistry 110: 36–44 Bouillon et al. (2014) Contrast- ing biogeochemical characteristics of the Oubangui River and tributaries (Congo River basin). Scientific Reports 4: 5402 | DOI: 10.1038/srep05402"

- We will add a short discussion as requested by both reviewers in Section 3.1 "DOM carbon dynamics" and Section 4.3 which will be renamed "Limitations, implications and future work"; this is outlined below:

To be added to Section 3.1: Based on the lack of a significant difference between DOC concentrations in the treatments at the initial time point (Figure 1), there is very little DOC contained between the 0.2 and 0.7 $\mu$m size range; this is supported by previous work (Zsolnay, 2003; Bouillon et al., 2014; Denis et al., 2017). Most DOM molecules are very small, less than $\sim$0.1 $\mu$m in size (Gustafsson and Gschwend, 1997), with very little in the 0.2 to 0.7 $\mu$m size range contributing to the overall DOC concentration. This is also reflected in the DOM quality indices in this study, although the P2 treat- ment shows some differences initially (Figure 2). The lower E2:E3 ratio and higher SR initial values for P2 compared to P7 suggests that the P2 samples are of lower molec- ular weight in general, while the higher SUVA254 values suggest greater aromaticity in the P2 samples (Helms et al., 2008). This difference is most clearly seen in the ab- sorbance at 440 nm (Figure 2e). However, these differences appear relatively minor in magnitude, with the structural indices converging by the first time point (day 5) across all treatments during incubation (Figure 2). This suggests that what small initial struc- tural differences existed between the P2 and P7 treatments appeared unimportant to the overall dynamics of the DOM pool, which is supported by the lack of difference in

initial DOC concentrations (Figure 1), and previous work (Zsolnay, 2003; Bouillon et al., 2014; Denis et al., 2017).

To be added to Section 4.3: This study suggests that filter size has relatively little im- pact on DOC concentration and structure, supporting previous studies (Zsolnay, 2003; Bouillon et al., 2014; Denis et al., 2017). The choice of filter size for DOM sample stor- age, therefore, is relatively unimportant. What is more important for sample storage, and the subsequent degradation of the DOM pool by latent microbes, is the imme- diate treatment and storage conditions. For example, acidification, storage in dark conditions, refrigeration and freezing can all reduce microbial activity and maintain the integrity of DOM samples, but the best method is to analyse the samples as soon as possible after collection (Gulliver et al., 2010; Peacock et al., 2015).

References:

Bouillon, S., Yambele, A., Gillikin, D. P., Teodoru, C., Darchambeau, F., Lambert, T., Borges, A.V.: Contrasting biogeochemical characteristics of the Oubangui River and tributaries (Congo River Basin), Sci. Rep., 4, 5402, doi:10.1038/srep05402, 2014.

Denis, M., Jeanneau, L., Pierson-Wickman, A.-C., Humbert, G., Petitjean, P., Jaffrezic, A., Gruau, G.: A comparative study on the pore-size and filter type effect on the molec- ular composition of soil and stream dissolved organic matter, Organic Geochemistry, 110, 36-44, doi:10.1016/j.orggeochem.2017.05.002, 2017.

Gulliver, P., Waldron, S., Scott, E. M., Bryant, C. L.: The effect of storage on the radio- carbon, stable carbon and nitrogen isotopic signatures and concentrations of riverine DOM, Radiocarbon, 52(2-3), 1113-1122, doi:10.1017/S0033822200046191, 2010.

Gustafsson, O., Gschwend, P. M.: Aquatic colloids: Concepts, definitions, and current challenges, Limnol. Oceanogr., 42(3), 519-528, doi:10.4319/lo.1997.42.3.0519, 1997.

Peacock, M., Freeman, C., Gauci, V., Lebron, I., Evans, C. D.: Investigations of freez- ing and cold storage for the analysis of peatland dissolved organic carbon (DOC)

and absorbance properties, Environ. Sci.: Processes Impacts, 17, 1290-1301, doi:10.1039/c5em00126a, 2015.

Zsolnay, A.: Dissolved organic matter: artefacts, definitions, and functions, Geoderma,113,187-209, doi:10.1016/S0016-7061(02)00361-0, 2003.

---

## Author Response (AR1)

To the Editor,

We thank you and the reviewers for taking the time to go through our manuscript. We have submitted a revised manuscript and include below a version showing the changes we've made highlighted in yellow.

Please find below our response to each of the reviewer comments in **red text**, with the page/line reference (e.g., "P.2, L.5-11") referring to the marked-changes manuscript. The reviewer comments were very helpful and have improved the manuscript, and we thank them for their contributions.

Many thanks for your consideration of our manuscript,
Joshua Dean
(On behalf of all co-authors)

--

**Reviewer #1**

This is overall a short but relevant manuscript, that provides experimental data on DOM degradation of surface waters under different filtration conditions.
We thank the reviewer for their positive and constructive comments.

"DOM carbon concentrations" is odd and confusing – I suggest to use "DOC concentrations"
We used DOM carbon concentrations so we didn't confuse readers by switching between the acronyms "DOM" and "DOC", but we acknowledge this is not the norm so we revert to using both acronyms (e.g. P.3, L.9).

The description of DOC temporal variations in the experiments (Figure 1) is perhaps not very convincing. Based on many published degradation experiments, one would expect to see exponential losses, the smoothing approach here is odd.
The reviewer is correct that it is more common to fit a negative exponential model to data of this type. However, this function is constrained to be strictly decreasing and therefore cannot be sensibly fit to data that show increasing values with time, or even lag-phase dynamics. Rather than impose a functional form on the data we therefore decided to use a smoothing approach (GAMs), combined with model selection methods to guard against overfitting. This approach has been successfully applied elsewhere (Catalan et al. 2017). We have also fit negative exponential models to the same data (resulting in much poorer fits, AIC = -13), and include this in the updated Supplementary Material and refer to it in the Results (P.6, L.18-21).

DOC concentrations were measured using a TOC-IRMS setup, it is unfortunate that d13C data are not presented, these could have a strong added value to the manuscript and would allow to expand the discussion with other published studies looking at changes in d13C during microbial degradation, or to shed some light on the suggestions that an initial (small) increase in DOC could be due to nitrification.
We initially excluded these results because there was very little change in the $\delta^{13}C$ values during the experiment. However, after exploring the literature further, and after the encouragement of the reviewer, we found that reporting $\delta^{13}C$-values during these incubations is very rare. We now include these results in the new Fig. 2, and discuss the results briefly in

the text (P. 6, L.28 to P.7, L.9).

It would be good to include some discussion/references on how the filter pore size might (or might not) affect the DOC concentrations and characteristics. As a starter, both papers below demonstrate no significant differences in DOC and d13C-DOC between 0.2 and 0.7 um filtrations: Denis et al. (2017) A comparative study on the poresize and filter type effect on the molecular composition of soil and stream dissolved organic matter. Organic Geochemistry 110: 36–44 Bouillon et al. (2014) Contrasting biogeochemical characteristics of the Oubangui River and tributaries (Congo River basin). Scientific Reports 4 : 5402 | DOI: 10.1038/srep05402

We have added a short discussion on the expected influence of filter pore size on DOC concentration and characteristics starting with the references as suggested (P.7, L.18-29). Please see our response to Reviewer 2's comment on this.

Throughout the ms, you refer to either 'filtering' or 'filtration' – I would prefer to stick to the latter

We have gone through the ms and checked for consistency throughout with reference to "filtration" (see for example in the title)

Figure 1: I do not see a legend to indicate which experiments the data correspond to.

We have corrected the missing labels in Figure 1.

--

**Reviewer #2**

Dean et al. present a concise method focused manuscript into the use of different filter (pore) sizes and the consequences of these on DOM degradation. The manuscript is well written and organised and easy to follow. I have the following comments:

We thank the reviewer for these positive comments and helpful feedback.

Methodology

Line 33- What temperature were the samples refrigerated at, would be useful to note? And were the samples cold stored on the journey back from the field to the University.

The samples were refrigerated at 4°C. The samples were not refrigerated during transport, but the travel time was less than an hour and were refrigerated immediately upon arrival at the university. These notes have been added to the methodology in the revised manuscript (P.3, L.1-3).

Was there a reason that only sizes 0.2 and 0.7um pore sizes were chosen? As 0.45um is also a very common size, particularly in tropical peatland studies.

We chose 0.2 and 0.7 μm pore sizes as these are the most commonly used filter sizes for DOM incubations and provide a reasonable spread in pore size. The reviewer is correct in that 0.45 μm is common as a size definition or filtration choice for normal DOC concentration analyses (as initially mentioned in our introduction, P.2, L.9). But 0.45 μm is not commonly used for incubations unless an inoculum is added, in which case we would consider a 0.45 μm treatment to be analogous to our 0.2 μm treatment. We did not have the budget for a fourth treatment in this experiment, but note this analogy in the methods (P.3, L.31-32).

Just wondered why it took 6 months to analyse the samples for their carbon concentration?
It took 6 months to analyse the samples because our own lab for running DOC concentrations had technical problems, so we had to find another lab to run these samples. This took some time and there was a wait to finally run the samples. 6 months frozen storage should not have had any significant impact on the concentration and isotopic composition of the DOC samples (Gulliver et al., 2010; Peacock et al., 2015). We have added a statement on this and included the appropriate references in the text (P.3, L.30-31).

Results

Figure 1. There is no indication of the legend to highlight which line/ colour refers to P2, P7 and UF.
We have corrected the missing labels.

Figure 2. Perhaps the marker points could be made a bit smaller, so it is easier to distinguish the individual points.
We have shrunk the marker points as requested.

Include P2, P7 and UF under the Day 0 and Day 14 in Figure 3c.
We have included the P2, P7 and UF labels under Day 0 and Day 14.

Discussion

It would be good to see more of a discussion concerning the impact of pore size on DOC concentration, and particularly how this may change over time. For example, would any pore size be fine if you are going to analyse the sample quickly?
This comment is echoed by Reviewer 1 (see above). We have added a short discussion as requested by both reviewers in Section 3.1 "DOC dynamics" (P.7, L.18-29) and Section 4.3 (P.11, L.7-12), which is renamed "Wider implications, limitations and future work". We also add a short paragraph referring to a relevant very recent paper on using a potential universal bacterial inoculum to help address some of the issues raised in our manuscript, which we feel adds to the context and depth of the discussion (Pastor et al., 2018).

--

**Filtration artefacts in bacterial community composition can affect the outcome of dissolved organic matter biolability assays**

Joshua F. Dean[1,2,], Jurgen R. van Hal[2], A. Johannes Dolman[1], Rien Aerts[2], and James T. Weedon[2]

[revised manuscript text omitted]

30     months, which should not have had any significant impact on the concentration or isotope composition of the DOC analyses (Gulliver et al., 2010; Peacock et al., 2015). 0.2 μm and 0.7 μm are the most commonly used filter sizes for DOM incubations (Vonk et al., 2015). 0.45 μm is also a common filter size for DOC sample collection, but less so for incubations.

DOC concentrations and $\delta^{13}$C-DOC isotopes were measured on a high-temperature combustion total organic carbon system (varioTOC cube; Elementar Analysensysteme GmbH) coupled to an isotope ratio mass spectrometer (IsoPrime 100; Isoprime Ltd, UK), using an independent calibration curve (measurement range 0.02 to 12.5 mmol/L) with $^{13}$C isotope standards IAEA-600, caffeine and IAEA-CH6, sucrose (Federherr et al., 2014; Kirkels et al., 2014).

[revised manuscript text omitted]
 1). It is generally more common to fit a negative exponential model to DOM biolability data. However, this function is strictly constrained to decreasing trends and cannot be sensibly fit to data that show increases, as seen in this study (Figure 1). We attempted to fit negative exponential models to the same data, but this resulted in a much poorer fit (AIC = -13; supplementary Figure S1). The difference in dynamics according to filtration strategy was statistically significant when analyzed with GAMs, since the best fitting model included separate smoothers for each filtration treatment (*adjusted $R^2$* = 0.89, estimated *d.f.* = 11.0, Table 1). Two points at UF day 41 could be considered outliers; however, removing them from the dataset leads to qualitatively the same conclusions with regards to the best fitting model by the AIC model comparison, and the following predicted values at day 27 (proportion DOC remaining, mean ± standard error, *n* = 5 in all cases; P2 = 98% ± 3, P7 = 27% ± 3, UF = 86% ± 3), and at day 41 (P2 = 31% ± 3, P7 = 21% ± 3, UF = 24% ± 4) (Table 1; Figure 1).

The $\delta^{13}$C-DOC values remained stable during incubation, within ± 2.0 ‰ (Figure 2), although this variability is slightly more pronounced than in other dark bacterial decomposition experiments of aquatic DOC (± 0.0-0.5 ‰; Lalonde et al., 2014; Vähätalo and Wetzel, 2008). The shifts in $\delta^{13}$C-DOC could be due to the preferential mineralization of certain DOC molecules with distinct $\delta^{13}$C signatures: a more positive $\delta^{13}$C signature for all samples at day 5, and a more negative $\delta^{13}$C signature in only the P7 samples at day 14. The former shift reflects the initial increase in DOC concentrations seen in the P2

and P7 treatments, and the latter shift reflects the change in DOC degradation dynamics seen only in the P7 treatment. As the experiment progressed, the $\delta^{13}C$ across all samples converge, although over a wide range (-28.0 to -29.0 ‰; Figure 1). DOC concentrations in P2 and P7 were slightly higher than initial measurements on day 5 (Figure 1), suggesting there may have been growth of chemoautotrophic organisms (photo-autotrophy can be ruled out given the dark incubations). A likely candidate is nitrification, given the relatively large concentrations of ammonia in the water column (29-82 µmol/L $NO_3$ + $NO_2$, J. van Huissteden, pers. comm.). This is supported by rapid growth in microbial cell counts over the same period (Logue et al., 2016) (Supplementary Figure S2). However, the shifts in $\delta^{13}C$-DOC do not provide any clear line of reasoning for these differing DOC dynamics (for example a shift towards less negative $\delta^{13}C$-DOC values due to chemoautotrophic fixation of ambient $CO_2$ at day 5).

DOM quality showed a consistent pattern across filtration treatments. All the absorption parameters indicate a rapid shift from more aromatic/higher molecular weight DOM to much more degraded and smaller molecules by day 5 (Figure 3). This is more pronounced in SUVA254, E2:E3 ratios and absorptions at 240, 300 and 440 nm compared to $S_R$, the latter of which shows greater scatter, but the same general trend. Moreover, factorial ANOVA analyses show that the temporal effect was always stronger than any effect of filtration treatment (Table 2). The E4:E6 ratios did not change significantly over the course of the experiment, nor did they differ between filtration strategy treatments. This shows that the DOC concentration dynamics are decoupled from DOM structural dynamics, indicating that microbial activity is playing an important role in DOC concentrations, but that overall molecular degradation was consistent across the treatments.

Based on the lack of a significant difference between DOC concentrations in the treatments at the initial time point (Figure 1), there is very little DOC contained between the 0.2 and 0.7 µm size range; this is supported by previous work (Zsolnay, 2003; Bouillon et al., 2014; Denis et al., 2017). Most DOM molecules are very small, less than ~0.1 µm in size (Gustafsson and Gschwend, 1997), with very little in the 0.2 to 0.7 µm size range contributing to the overall DOC concentration. This is also reflected in the DOM quality indices in this study, although the P2 treatment shows some differences initially (Figure 3). The lower E2:E3 ratio and higher $S_R$ initial values for P2 compared to P7 suggests that the P2 samples are of lower molecular weight in general, while the higher SUVA254 values suggest greater aromaticity in the P2 samples (Helms et al., 2008). This difference is most clearly seen in the absorbance at 440 nm (Figure 3e). However, these differences appear relatively minor in magnitude, with the structural indices converging by the first time point (day 5) across all treatments during incubation (Figure 3). This suggests that what small initial structural differences existed between the P2 and P7 treatments appeared unimportant to the overall dynamics of the DOM pool, which is supported by the lack of difference in initial DOC concentrations (Figure 1), and previous work (Zsolnay, 2003; Bouillon et al., 2014; Denis et al., 2017).

[revised manuscript text omitted]

**4.3 Wider implications, limitations and future work**

Using a universal bacterial inoculum could prevent the biolability comparison issues raised by this study (Pastor et al., 2018). However, such a universal inoculum could also suffer from changes in microbial community composition over time depending on the DOM source and available nutrients, and/or be affected by microbes that slip through a 0.2 μm filter (Hahn, 2004). Further trials of universal inoculums across a range of aquatic environments would be beneficial in assessing these potential issues.

This study suggests that filter size has relatively little impact on DOC concentration and DOM structure, supporting previous studies (Zsolnay, 2003; Bouillon et al., 2014; Denis et al., 2017). The choice of filter size for DOM sample storage, therefore, is relatively unimportant. What is more important for sample storage, and the subsequent degradation of the DOM pool by latent microbes, is the immediate treatment and storage conditions. For example, acidification, storage in dark conditions, refrigeration and freezing can all reduce microbial activity and maintain the integrity of DOM samples, but the best method is to analyze the samples as soon as possible after collection (Gulliver et al., 2010; Peacock et al., 2015).

This study is from a single peatland site in the Netherlands and therefore may not be representative of DOM biolability dynamics across a wider range of freshwater ecosystems. The organic carbon-rich nature of the study site and its water are consistent with temperate peatland characteristics elsewhere in the Northern Hemisphere (e.g. Billett et al. 2010), but the agricultural history of the site means that nutrient availability and the general chemistry of the sample waters are likely different from more pristine peatland systems. Such environmental factors are important influences on microbial communities (Dean et al., 2018; Gulis and Suberkropp, 2003), which would have influenced the development of bacterial community composition during the experiment presented here.

It is important to note that the microbial community patterns described here are based on relative abundances; thus, apparent declines of particular taxa may be attributable to either real declines in population, or (as is more likely) relatively slower growth relative to other taxa in the community. Moreover, PCR amplicon generation is known to introduce biases in the community profile (e.g. Sipos et al. 2007), so any interpretation of these data for particular taxa should be further confirmed by more reliable quantitative methods such as taxon-specific qPCR (Smith and Osborn, 2009), FISH (Amann et al., 1995) or PCR-free metagenomics methods (Handelsman, 2004).

Despite the lack of clear connection between bacterial community composition and DOM dynamics, the high reproducibility of both measurements across the technical replicates suggest that this system could be a potentially useful model for further exploring the relationship between community structure and function. A central question in microbial ecology is to what extent information about the taxonomic composition of microbial assemblages allows prediction of their biogeochemical function (Bier et al., 2015; Graham et al., 2016). Future studies should seek to experimentally test the relative importance of initial population size and the presence or absence of grazers in determining DOM utilization rates. Future work should also consider bacterial community composition dynamics, size selection and DOM structural characteristics across multiple sites

and climate settings to confirm the degree to which filtration strategy affects DOM biolability assays across a wider range of catchment settings.

**5 Conclusions**

Filtration strategy was shown in this study to affect both microbial community composition and DOM degradation dynamics, but these two responses were disconnected from one another. There are two important conclusions from these results for interpreting aquatic biolability assays. Firstly, our results suggest that care should be taken when comparing results using different filtration strategies, especially for shorter incubations since we have shown that the DOM degradation and microbial community dynamics may not converge until after the commonly used 28-day length for long-term biolability assays, if at all (Figures 1 and 4). However, in many cases the relevant parameter to be estimated from such long-term incubations is the total 'biolabile fraction' (Guillemette and del Giorgio, 2011), and despite the divergent dynamics in the early part of our incubation it appears that all filtration strategies do eventually converge on a similar value for total DOM biolability. The P7 (0.7 μm filtration with no inoculum) treatment appeared to reach this point at the fastest rate.

**Acknowledgements**

This work was carried out under the program of the Netherlands Earth System Science Centre (NESSC), financially supported by the Ministry of Education, Culture and Science (OCW) (grant number 024.002.001). We thank the Systems Bioinformatics group at VU Amsterdam for the use of their flow cytometry equipment. We also thank the editor, Prof Gerhard Herndl, and two anonymous referees for their comments and discussion which have improved this manuscript. The authors declare that they have no conflict of interest.

[revised manuscript text omitted]